# The Influence of Red Meat on Colorectal Cancer Occurrence Is Dependent on the Genetic Polymorphisms of S-Glutathione Transferase Genes

**DOI:** 10.3390/nu11071682

**Published:** 2019-07-22

**Authors:** Justyna Klusek, Anna Nasierowska-Guttmejer, Artur Kowalik, Iwona Wawrzycka, Magdalena Chrapek, Piotr Lewitowicz, Agnieszka Radowicz-Chil, Jolanta Klusek, Stanisław Głuszek

**Affiliations:** 1Department of Surgery and Surgical Nursery with a Research Laboratory and Genetic Laboratory, Faculty of Medicine and Health Sciences, Jan Kochanowski University, 19, 5-317 Kielce, Poland; 2Department of Pathology, Faculty of Medicine and Health Sciences, Jan Kochanowski University, 19, 25-317 Kielce, Poland; 3Department of Molecular Diagnostic, Holy Cross Cancer Centre, 19, 25-734 Kielce, Poland; 4Department of General, Oncological and Endocrinological Surgery, Voivodeship Hospital, 19, 25-736 Kielce, Poland; 5Department of Probability Calculus and Statistics, Institute of Mathematics, Jan Kochanowski University, 19, 25-406 Kielce, Poland; 6Department of Physiology, Institute of Biology, Jan Kochanowski University, 19, 25-406 Kielce, Poland

**Keywords:** red meat, colorectal cancer, *GSTT1*, *GSTM1*, *GSTP1*

## Abstract

Background: It is postulated that both individual genotype and environmental factors such as diet may modify the risk of developing colorectal cancer (CRC). The influences of *GST* gene polymorphism and red meat intake on CRC occurrence in the Polish population were analyzed in this study. Methods: Genotyping was performed with the qPCR method. Results: A high frequency of meat consumption was associated with an over 2-fold increase in the risk of colorectal cancer odds ratio (OR) adjusted for sex and age = 2.4, 95% confidence interval (CI); 1.3–4.4). However, after analyzing the genetic profiles, in the absence of polymorphisms of all three analyzed genes, there was no association between a high frequency of meat consumption and the occurrence of CRC. In the case of *GSTM1* gene polymorphism, the high frequency of meat consumption increased the risk of CRC by almost more than 4 times (OR adjusted for sex and age = 3.8, 95% CI: 1.6–9.1). For *GSTP1* gene polymorphism, a 3-fold increase in CRC risk was observed with a high frequency of meat consumption (OR adjusted for sex and age = 3.4, 95% CI: 1.4–8.1). In the case of *GSTT1* gene polymorphism, the increase in risk of CRC was not statistically significant (OR adjusted for sex and age = 1.9, 95% CI: 0.4–8.5). Conclusions: The frequency of red meat intake in non-smokers increases the risk of colon cancer in the case of *GST* gene polymorphisms.

## 1. Introduction

Colorectal cancer (CRC) is the third most commonly diagnosed cancer and the second most deadly cancer worldwide. According to GLOBOCAN 2018 data, there were 1.8 million new cases of CRC and nearly 1 million deaths due to this disease in 2018 [1]. The etiology of colorectal cancer is the subject of genetic and epidemiological studies worldwide. It is postulated that both polymorphisms of low penetration genes and environmental factors such as diet and broadly understood lifestyle factors may modify the risk of developing this type of cancer. The importance of environmental factors in the risk of cancer development is so significant that the National Cancer Institute estimates that almost 70% of cases could have been prevented [2].

A high percentage of colorectal cancer cases are found in developed countries with a high degree of industrialization, where the society is characterized by a so-called Western lifestyle. A diet rich in red processed meat seems to be an integral element of this lifestyle [3,4]. Thus, it is considered one of the environmental factors promoting the development of CRC. In Poland, the consumption of red meat increased 2-fold in the period 1960–1975, but in the 1990s, it decreased, and subsequently, despite some fluctuations, it has remained basically on a constant level of about 45 kg per year [5].

The mechanism underlying the potential carcinogenicity of red meat has not been entirely explained so far. Here, several potential mechanisms are postulated. Most attention is focused on heterocyclic amines (HAA) as well as polycyclic aromatic hydrocarbons (PAH), which arise during the thermal processing of red meat [6]. These compounds are classified into category 2 by the International Agency for Research on Cancer and are referred to as probably carcinogenic to humans [3]. Moreover, the genotoxicity of these compounds has been confirmed in research on bacterial and mammalian cells in vitro, and it has also been demonstrated that they induce colon tumors in rats [2].

It is noteworthy that in processed meat with a high content of salts, nitrates, and nitrites, there are N-nitroso compounds with mutagenic and carcinogenic potential. In addition, these compounds are endogenously formed as a result of meat digestion [3]. A high level of heme iron in red meat is catalytically important for the formation reaction of harmful nitroso compounds during its passage through the digestive tract. Furthermore, heme iron promotes the formation of reactive oxygen species (ROS) with strong mutagenic properties. ROS exhibit cytotoxic effects and cause inflammation, which, in turn, promotes epithelial cell hyperproliferation [7].

In a prospective study of the American population by the National Institutes of Health (NIH) including around 0.5 million people, it was confirmed that more frequent red meat consumption increases the risk of CRC; however, this relationship was stronger for rectal cancer than for colon cancer [8]. A meta-analysis of 22 studies from around the world showed a strong positive relationship between red meat consumption more than once a day and cancer of both the colon and rectum [4].

Despite the fact that red meat has been recognized as a cancer-promoting factor, the results of studies do not always explicitly confirm the relationship between the consumption of red and processed meat with the risk of gastrointestinal cancers [6,9]. A pooled analysis of over 8000 CRC cases and more than 9000 controls did not indicate a significant relationship between red meat consumption and colorectal cancer in prospective studies, but only for case–control studies [10]. In addition, similar results were obtained in a meta-analysis of studies on the association between red meat consumption and gastric cancer. A positive relationship was found for case–control studies, however, not for prospective studies [11].

The lack of unequivocal conclusions may result from different types and patterns of research or from the limitations of experiments analyzing the frequency of consumption (food frequency questionnaires (FFQs)) [6]. Perhaps it may be important to associate environmental factors, including the supply of red meat in the diet, with genetic predisposition in patients with an increased risk of developing CRC. The genes for S-glutathione transferase (*GST*), which are responsible for detoxification processes, including antioxidant activity, belong to the group of genes whose polymorphisms may be crucial for carcinogenesis. They represent a superfamily of genes that encode phase II metabolizing enzymes for a number of toxic substances. Their substrates are, among others, heterocyclic amines, which are potential carcinogens in red meat subjected to thermal treatment. In addition, the polymorphism of *GST* genes results in a decrease or a complete lack of enzymatic activity of coded transferases [12]. Together with the simultaneous high consumption of red meat, a deficiency of GST enzymes may cause increased exposure of the intestinal mucosa to toxins such as HAA, PAH, N-nitrosamines, and ROS, which may be conducive to cancer-forming processes.

Based on this hypothesis, the aim of the study was to analyze the correlation between *GST* gene polymorphism and the frequency of red meat consumption in the risk of developing colorectal cancer in a Polish population. Confirmation of possible dependencies between the high consumption of red meat in people with decreased detoxification activity resulting from *GST* gene polymorphism and the risk of developing CRC would allow better future adaptation of prophylactic recommendations to this group of patients.

## 2. Materials and Methods

### 2.1. Study Population

The recruitment of patients participating in the study took place at two clinical centers in Kielce, Poland, the Provincial Hospital in Kielce and the Holy Crosss Cancer Centre, in 2014–2017. Current and former smokers were excluded from the population. The same carcinogenic compounds are contained in tobacco smoke that arises from the heat treatment process and/or the passage of red meat through the digestive tract. Thus, smoking by patients disrupts the assessment of the impact of red meat intake in the context of exposure to carcinogenic substances such as HAA, PAH, ROS, and N-nitroso compounds. The study group included 197 patients with confirmed colorectal cancer based on a pathomorphological diagnosis of specimens collected during colonoscopy or surgery. The control group consisted of 104 patients without cancer, as confirmed by an endoscopic and/or histopathological examination. All patients signed written consent forms prior to participation in the study. The study was approved on 3 June 2013 by the local Bioethics Commission (No. 5/2013) on the basis of the submitted application with an exact description of the procedure.

### 2.2. Dietary Assessment

The frequency of red meat consumption was assessed with the FFQ. The questions concerned the usual frequency of red meat consumption, including the consumption of beef, pork, veal, mutton, and venison, categorized by the method of preparation. It is noteworthy that each question covered one kind of meat dish, without differentiation for particular type of red meat (e.g., chop, stew, roast, meatballs, and others) to be as respondent friendly as possible. Questions 1–5 concerned the frequency of consumption of red meat subjected to thermal treatment (roasting, frying, braising, grilling), whereas questions 6–8 concerned the frequency of processed red meat consumption in the form of cold meat, canned food, and sausages. The questionnaire was completed independently by the patients after prior instructions from a nurse trained to participate in the project.

### 2.3. Genotyping

Peripheral blood leukocytes were used for genetic testing. DNA was isolated from frozen blood samples using the Genomic Micro AX Blood Gravity kit from AA Biotechnology (Gdynia, Poland). The purity and concentration of isolated DNA were evaluated spectrophotometrically at 260 nm and 280 nm (NanoDrop 2000, Thermo Fisher Scientific,) Waltham, MA, USA). Analysis of the *GSTT1* (Assay ID Hs00010004_cn) and *GSTM1* (Assay ID Hs02575461_cn) was conducted using the qPCR relative quantification method. The *TERT* gene constituted the internal control of reaction. Analysis of the *GSTP1* gene SNP (rs1695) polymorphism was conducted using TaqMan qPCR endpoint genotyping (Assay ID C_3237198_20). In all cases, the Light Cycler 96 instrument and TaqMan primer/probe kit (produced by Life Technology, Carlsbad, CA, USA) were used.

### 2.4. Statistical Analysis

Categorical data were expressed as number and percentage distributions. The chi-squared test or Fisher’s exact test were applied to compare proportions. Numerical variables were presented as the median and interquartile range and compared by the Mann–Whitney *U* test due to non-normality which was assessed by the Shapiro–Wilk test. Crude and adjusted odds ratios (OR) with 95% confidence intervals (95% CI) were calculated in a logistic regression model. A two tailed *p*-value < 0.05 was considered statistically significant. All statistical analyses were performed using R (version 3.1.2; The R Foundation for Statistical Computing, Vienna, Austria) and Statistica (StatSoft, Inc. 2014, version 12 Cracow, Poland.

## 3. Results

A total of 301 non-smokers of both sexes, aged 38 to 81, were genotyped and interviewed. The demographic characteristics of the studied population were presented in our previous study [13]. The vast majority of cancers were adenocarcinoma grade G2. The distribution of cancer localization between the colon and the rectum was fairly even (51.8% vs. 44.2%).

In both groups (patients and controls), the distribution of polymorphisms in *GST* genes was analyzed. There was no statistically significant relationship between the occurrence of polymorphism in the individual genes studied and CRC, as was reported in our earlier publication [13]. Additionally, the frequency of meat consumption was analyzed in terms of low (not more than 6 times a week) and high (7 times a week or more often) (Table 1) on the basis of the reports of other investigators [4,14]. Our further studies showed that the consumption of red meat (mainly beef and pork) in the form of cooked, roasted, and fried dishes as well as various types of cold meat and offal in a Polish population is very high. As many as 81.1% of all people participating in the study declared daily or more frequent consumption of this product group. As shown in Table 1, the frequency of red meat consumption varied between patients and controls. Moreover, it was noticed that regardless of the genetic profile, many more individuals with colorectal cancer reported eating red meat more than 7 times a week (85.8%), where this high level of consumption was noted in 72.1% of controls. As shown in Table 1, a high frequency of meat intake was associated with a statistically significant increase in the risk of colon cancer of over 2-fold (crude OR = 2.3 and OR adjusted for sex and age = 2.4) (Table 1).

The study examined the correlation between the frequency of red meat consumption and the *GST* gene polymorphism in the risk of developing CRC. The results of this analysis are presented in Table 2. There was no association between a high frequency of meat consumption and the occurrence of CRC in the absence of polymorphisms of all three studied genes. If a polymorphism of at least one of the three analyzed genes occurred, a high frequency of meat consumption increased the risk of CRC by about 3 times. In the case of the *GSTM1* gene polymorphism, a high frequency of meat consumption increased the risk of CRC by almost 4 times. Furthermore, for polymorphism of the *GSTP1* gene, a 3-fold increase in CRC risk was observed with a high frequency of meat intake. In the case of the *GSTT1* gene polymorphism, a high frequency of meat consumption increased the chance of CRC occurring in a statistically insignificant manner, which may have resulted from the small size of the group (52 people) (Table 2).

## 4. Discussion

Cytosolic glutathione S-transferase genes seem to be highly polymorphic in humans. They occur in dozens up to hundreds of variants observed in populations around the world with varying frequency. Polymorphisms with a high or even 50% frequency of occurrence in Europe were analyzed in the study. The frequency of analyzed polymorphisms obtained for the Polish population confirmed the data acquired by other researchers from Europe [13,15,16].

The analyzed polymorphisms reduce the detoxification capacity of this group of enzymes in theory, but, in fact, there was no statistically significant relationship between the genetic profile of individuals and the risk of developing CRC. Therefore, the aim of the project was to investigate the interaction between genetic factors and the influence of the environment in the form of specific eating habits in modulating the risk of colon cancer. Red and processed meats seem to be elements of the diet with highly unfavorable effects on the intestinal mucosa of people with reduced detoxification potential. The International Agency for Research on Cancer (IARC), a member of the World Health Organization (WHO) within the European Code Against Cancer, recommends reducing the consumption of red and processed meat in the diet, especially due to the risk of CRC [17].

According to Jarosz and coauthors, the consumption of this product group was the highest in Poland in the late 1970s and early 1990s, which could have influenced currently diagnosed cancer cases [5]. It is postulated that long-term meat consumption, rather than acute intake, can accelerate CRC [18]. There are some data reporting the approximate time of red meat consumption needed to have an impact on cancer development. In a pooled analysis, Bernstein and coauthors found that the risk of CRC increases after a lag of 4–8 years or more from the time of red meat consumption [14]. The current frequency of red meat consumption in the Polish population was analyzed in the project. This consumption appears to be at a very high level, because less than 19% of respondents declared an intake less frequent than daily. According to some authors, consumption of red meat at least 3 times a week may increase the risk of CRC by as much as 80% [19]. Interestingly, only 4.32% of people in the studied population declared consumption below this level. According to a meta-analysis of nine studies, a frequency of red meat consumption higher than daily significantly increases the risk of developing both colon and rectal cancers [4]. The results of our research are consistent with this meta-analysis (Table 1).

A statistically significant difference (*p* = 0.005) between the patients and controls was presented in terms of the number of people declaring a high consumption of red meat (85.8% vs. 72.1%, respectively). Based on the above data, it can be assumed that very frequent consumption of red meat may potentially increase the risk of developing colorectal cancer. The probable mechanism conducive to carcinogenesis may be the regular exposure of mucous membrane to toxins generated in the digestion process of this food product, such as HAA, nitrosamines, ROS, or increased oxidative stress resulting from a high content of heme iron [3,4]. Nevertheless, after analyzing the genetic profiles in terms of the polymorphism of selected *GST* genes, it was found that such a risk increases only in people with a deficiency of glutathione S-transferases resulting from mutations of the genes encoding them.

Statistically significant differences in the frequency of red meat intake between the patients and controls were observed for people with polymorphism of *GSTM1* (*p* = 0.003) and *GSTP1* (*p* = 0.018) genes. The studied polymorphism of the *GSTM1* gene is based on a deletion of both gene alleles (genotype null/null) and results in a lack of protein product, so it can be assumed that deficiency of this detoxification enzyme glutathione S-transferase µ1 in connection with increased exposure to carcinogenic substances of red meat is a risk factor for CRC development.

According to our results (Table 2), in the case of the *GSTM1* gene polymorphism, a high frequency of meat consumption resulted in the highest risk of the three analyzed genes, almost a 4-fold increase in the risk of CRC. In fact, it is believed that glutathione transferase μ1 shows higher efficiency in the metabolism of xenobiotics than other *GST* enzymes and plays a key protective role against cytotoxic and genotoxic compounds in the gastrointestinal tract [16].

The analyzed polymorphism of the *GSTP1* gene (SNP 105 Ile/Val) resulted in the expression of a protein with reduced enzymatic activity. An amino acid change at position 105 of the polynucleotide chain from isoleucine to valine affects the geometry of the substrate binding area by *GSTP1* transferase [13]. This modification has resulted in an average 3-fold reduction in enzyme–substrate affinity in in vitro studies [10]. Our results showed that this polymorphism, in combination with regular and frequent (more than once per day) consumption of red meat, increases the risk of colorectal cancer by more than 3 times.

Analysis of the frequency of red meat intake in people without the *GST* polymorphism did not show statistically significant differences between patients and controls, which means that high red meat intake in people with a normal enzyme apparatus of the glutathione S-transferase family does not affect the risk of colon cancer. It can be assumed that despite the regular consumption of red meat, exposure to carcinogens such as HAA, N-nitroso compounds, and ROS is minimized thanks to efficient detoxification systems with the participation of GST transferases. 

There are some limitations in our study. The FFQ survey used to assess the frequency of consumption of red meat did not allow a quantitative assessment of the consumption of red meat by patients. Nevertheless, a meta-analysis of studies from Europe, North America, and Australia has shown that the regularity and high frequency of eating red meat is more important for the risk of developing CRC than the daily amount [4]. There was also a memory bias that we could not exclude, but to minimize it, the questionnaire only asked about red meat consumption in the last year. The strength of our study is that we excluded tobacco smokers to eliminate smoking as a confounding factor, but we realize that there are still many other genetic and environmental factors that could have impacted our results due to the very complicated pathogenesis of CRC. The recruitment of patients was significantly prolonged due to the exclusion of current or former smokers. About 60% of colorectal cancer patients were rejected due to this exclusion criterion, so the pool of patients was severely limited. The most important strength of our study is that it is the first study on this topic conducted in Poland to combine the genetic and epidemiologic risk factors for colorectal cancer.

## 5. Conclusions

There are some clear conclusions from the study. First of all, the red meat intake in the Polish population is very high. The study showed an increased risk of CRC in people with a *GST* gene polymorphism and high (over once per day) consumption of red and processed meat. Such features are exhibited by as many as 69.1% of controls without changes in the large bowel. It is therefore considered that this portion of the controls should have constant monitoring. They have a 3-fold higher than normal risk of developing colorectal cancer in the future. It is advisable to include those people in further prospective research on the subject, which will be the goal of project continuation.

## Figures and Tables

**Table 1 nutrients-11-01682-t001:** Frequency of red meat intake in groups of patients and controls. OR, odds ratio; CI, confidence interval.

Red Meat Intake	Patients (197)	Controls (104)	OR (95% CI); *p*-Value	
Low (no more than 6 times a week)	28 (14.2%)	29 (27.9%)	Reference	Reference
High (7 times a week or more often)	169 (85.8%)	75 (72.1%)	2.3 (1.3–4.2); 0.005	2.4 (1.3–4.4); 0.005

**Table 2 nutrients-11-01682-t002:** Relationship between the frequency of red meat intake and the status of colorectal cancer (CRC) depending on the occurrence of polymorphism in the examined genes.

*GST* Genotype	Red Meat Intake *	Patients	Controls	OR (95% CI); *p*-Value	
				Crude	Adjusted **
Wild type	low	7/42 (16.7%)	4/23 (17.4%)	Ref. level	Ref. level
	high	35/42 (83.3%)	19/23 (82.6%)	1.1 (0.3–4.1); 0.95	0.9 (0.2–3.8); 0.93
*GSTM1* or *GSTP1* or *GSTT1* polymorphism	low	21/155 (13.5%)	25/81 (30.9%)	Ref. level	Ref. level
	high	134/155 (86.5%)	56/81 (69.1%)	2.8 (1.5–5.5); 0.002	3.1 (1.6–6.3); 0.001
*GSTM1* polymorphism	low	12/92 (13.0%)	17/47 (36.2%)	Ref. level	Ref. level
	high	80/92 (87.0%)	30/47 (63.8%)	3.8 (1.6–8.3); 0.002	3.8 (1.6–9.1); 0.003
*GSTP1* polymorphism	low	15/114 (13.2%)	16/54 (29.6%)	Ref. level	Ref. level
	high	99/114 (86.8%)	38/54 (70.4%)	2.8 (1.3–6.2); 0.02	3.4 (1.4–8.1); 0.006
*GSTT1* polymorphism	low	5/31 (16.1%)	5/21 (23.8%)	Ref. level	Ref. level
	high	26/31 (83.9%)	16/21 (76.2%)	1.6 (0.4–6.5); 0.49	1.9 (0.4–8.5); 0.40

* low means consumed not more than 6 times a week, whereas high means 7 times a week or more often ** for age and sex.

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
