# Peer review of "The Influence of Red Meat on Colorectal Cancer Occurrence Is Dependent on the Genetic Polymorphisms of S-Glutathione Transferase Genes"

_nutrients, 2019, doi:10.3390/nu11071682_

Reviewer 1 Report

The manuscript describes GST gene polymorphisms are involved in red meat intake effects on the occurrence of colorectal cancer (CRC) in a Polish population. Of note, the authors found that in the absence of polymorphisms of 3 GST genes (GSTM1 wild, GSTT1 wild and GSTP1 Ile/Ile), there were no association between the high frequency of red meat consumption and the occurrence of CRC. However, the following points should be resolved;

1) The title of the paper does not represent the content. In the manuscript, the authors have shown us how the genetic background influences the increasing effect of red meat intake on the occurrence of CRC. The title should be changed.

2) (lines 25 to 27) No description about the correlation between the GST gene polymorphisms and the frequency of red meat consumption in the manuscript.

3) (lines 39 to 40) The first sentence in Conclusions should be deleted, because no evidence was presented in this manuscript.

4) (line 167) ‘72.1% of patients’ should be ’72.1% of control.’

5) Table 1 is not required, because this is exactly same as Table 1 in reference (8). The authors should refer the reference (8).

6)(lines 182 to 183) Although there is a description of “The distribution of polymorphisms of the examined genes” in a title of Table 2, there is no data on polymorphism in the contents of the table.

7) On Table 2, the authors divided the frequency of red meat consumption on the basis of seven times a week. What is the basis for that?

8) On Table 3, have the authors analyzed a group of people with both GSTM1 and GSTT1 polymorphisms?

 Author Response

1)     The title of the paper does not represent the content. In the manuscript, the authors have shown us how the genetic background influences the increasing effect of red meat intake on the occurrence of CRC. The title should be changed.

Answer: Title changed to:

„Red meat influence on colorectal cancer occurrence is dependent on genetic polymorphism of the S-glutathione transferases genes”

 2)     (lines 25 to 27) No description about the correlation between the GST gene polymorphisms and the frequency of red meat consumption in the manuscript.

Answer: The sentence changed to:

“The influence of GST genes polymorphism and red meat intake on CRC occurrence in the Polish population was analyzed” (in revised manuscript lines: 31-33) 

 3)     (lines 39 to 40) The first sentence in Conclusions should be deleted, because no evidence was presented in this manuscript.

Answer: The sentence deleted.

 4)     (line 167) ‘72.1% of patients’ should be ’72.1% of control.’

Answer: Expression changed (in revised manuscript line: 178) 

 5)     Table 1 is not required, because this is exactly same as Table 1 in reference (8). The authors should refer the reference (8).

Answer: Table 1 deleted, all consequent changes in following text made (for example numbers of following tables), references changed to our previous study as the Reviewer suggested (in revised manuscript reference

 6)     (lines 182 to 183) Although there is a description of “The distribution of polymorphisms of the examined genes” in a title of Table 2, there is no data on polymorphism in the contents of the table.

Answer: The title of Table 2 (in revised manuscript Table 1) changed to: “The frequency of red meat intake in groups of patients and controls”

 7)     On Table 2, the authors divided the frequency of red meat consumption on the basis of seven times a week. What is the basis for that?

Answer: The basis of division for high and low red meat intake, has been made due to Smolińska and Paluszkiewicz, 2010 (References 15 in revised manuscript). According to this metanalysis (15)  increasing risk for cancerogenesis is caused by red meat intake at least once a day or more often, which is equivalent to 7 times a week or more often. The explanation has been included in the Results section (lines 169-170 in revised manuscript) with additional reference (ref:3 in revised manuscript).

 8)     On Table 3, have the authors analyzed a group of people with both GSTM1 and GSTT1 polymorphisms?

Answer: We have not performed analysis in groups of GSTM1 null/null and GSTT1 null/null genotypes because of a very limited  numbers of subjects showing this combined polymorphism. In details it was 18 patients in group with CRC and 10 of controls. If divided them into subgroups of low and high red meat intake it would be only a few subjects in every subgroup.

Reviewer 2 Report

General: The authors have identified an interesting research question. “Nutritional factors modulating the risk of colorectal cancer in people with polymorphism of the S-glutathione transferase genes” is an interesting topic.

 The title is appropriate.

Abstract:

Line 39-40 Conclusion section: authors mention “ Red meat intake in Poland is higher than in other European populations” This is a generalized statement and I don’t think is a conclusion from this study and probably should not belong here in this section.

Introduction:

In the first paragraph please include the incidence and mortality rates for pancreatic cancer. Please include the reference https://doi.org/10.5114/pg.2018.81072

Line 52-53:  The sentence in unclear and needs to be modified. I am not sure whether the authors here are referring to the developed or developing countries?

Line 54: “be its integral element …” element of what?? Please mention it as sentence is incomplete.

Line 56: “development of RJG.” Might be a misprint.

Line 76 “RJG” please use correct abbreviation

Line 83: “RJG” please change it to correct abbreviation

 Methods:

In dietary assessment: authors mention that “The frequency of red meat consumption was assessed with the FFQ (food frequency questionnaire) questionnaire”. Please explain how did the researchers exclude the memory bias of the participants? If not please add it to the limitations of the study.

Results:

Can authors classify results based on the type of red meat taken?

What all red meat were included in the study…this should be mentioned in the methods section

Discussion Section:

Lines 205-207 authors mentions “According to Jarosz and coauthors the consumption of this product group was the highest in Poland in the late 1970s and early 90s, which could have influenced currently diagnosed cancer cases (Jarosz et al., 2013).” Is there any data to say how long after consumption and after how many years of consumption can cause CRC…please include relevant studies?

Authors need to discuss in detail about the limitations of the study. Especially including comments from above as mentioned in methods.

 Last paragraph can be made into a separate conclusion section.

Tables are appropriate.

English and grammar needs to be thoroughly checked by a native English speaker as I did see many grammatical and sentence formation errors.

Overall a well conducted study.

 Author Response

Abstract:

Line 39-40 Conclusion section: authors mention “ Red meat intake in Poland is higher than in other European populations” This is a generalized statement and I don’t think is a conclusion from this study and probably should not belong here in this section.

Answer: The statement has been deleted

Introduction:

In the first paragraph please include the incidence and mortality rates for pancreatic cancer. Please include the reference https://doi.org/10.5114/pg.2018.81072

Answer: The reference has been included in the first paragraph, lines in revised manuscript: 51-53

Line 52-53:  The sentence in unclear and needs to be modified. I am not sure whether the authors here are referring to the developed or developing countries?

Answer: The sentence has been corrected. Indeed the statement refers to developed countries,  lines in revised manuscript: 59-60

Line 54: “be its integral element …” element of what?? Please mention it as sentence is incomplete.

Answer: The sentence has been modified to be clear, line in revised manuscript: 61

Line 56: “development of RJG.” Might be a misprint.

Answer: misprint corrected, line in revised manuscript:63

Line 76 “RJG” please use correct abbreviation

Answer: misprint corrected, line in revised manuscript:83

Line 83: “RJG” please change it to correct abbreviation

Answer: misprint corrected, line in revised manuscript:90

Methods:

In dietary assessment: authors mention that “The frequency of red meat consumption was assessed with the FFQ (food frequency questionnaire) questionnaire”. Please explain how did the researchers exclude the memory bias of the participants? If not please add it to the limitations of the study.

Answer: Unfortunately there was no possibility to exclude the memory bias, the questionnaire comprises only last year of consumption to minimize the memory bias. Appropriate statement was added to the limitations of the study, lines in revised manuscript: 272-274

Results:

Can authors classify results based on the type of red meat taken?

Answer: Unfortunately it is not possible to classify the results based on the type of red meat,  because in the questionnaire, different types of meat were included into one question.  The reason is that numerous responders were unable to recognize what type of red meat the dish contained. That is why we have decided to ask  for example for frequency of consumption of stew or chop or other meat dish, without differentiation individual types of red meat. Questionnaire was designed to be as much respondent-friendly as it is possible.

What all red meat were included in the study…this should be mentioned in the methods section

Answer: There were included all kind of red meat dishes consumed in Poland. There were following types of red meat: beef, veal, pork, mutton and venison, categorized by method of preparation. Description provided in the subsection 2.2 Dietary Assesment has been extended, lines:131-140.

Discussion Section:

Lines 205-207 authors mentions “According to Jarosz and coauthors the consumption of this product group was the highest in Poland in the late 1970s and early 90s, which could have influenced currently diagnosed cancer cases (Jarosz et al., 2013).” Is there any data to say how long after consumption and after how many years of consumption can cause CRC…please include relevant studies?

Answer: Relevant studies included in Disscussion, in revised manuscript lines:219-223, ref: 3,4

Authors need to discuss in detail about the limitations of the study. Especially including comments from above as mentioned in methods.

Answer: The “limitations of the study” have been expanded due to Reviewer comments and  made into separate section.

Last paragraph can be made into a separate conclusion section.

Answer: It has bee done

Tables are appropriate.

English and grammar needs to be thoroughly checked by a native English speaker as I did see many grammatical and sentence formation errors.

Answer: The manuscript has been corrected by the MDPI English editing Service

Overall a well conducted study.

Round  2

Reviewer 1 Report

I have no further comments.

Reviewer 2 Report

authors have made significant improvements to the manuscript and have made changes as per the reviewer recommendations. Manuscript looks complete now. Thank you for giving me the opportunity to review this manuscript.